# Treatment-Emergent Cilgavimab Resistance Was Uncommon in Vaccinated Omicron BA.4/5 Outpatients

**DOI:** 10.3390/biom13101538

**Published:** 2023-10-18

**Authors:** Cesare Ernesto Maria Gruber, Fabio Giovanni Tucci, Martina Rueca, Valentina Mazzotta, Giulia Gramigna, Alessandra Vergori, Lavinia Fabeni, Giulia Berno, Emanuela Giombini, Ornella Butera, Daniele Focosi, Ingrid Guarnetti Prandi, Giovanni Chillemi, Emanuele Nicastri, Francesco Vaia, Enrico Girardi, Andrea Antinori, Fabrizio Maggi

**Affiliations:** 1Laboratory of Virology, National Institute for Infectious Diseases “Lazzaro Spallanzani” (IRCCS), 00149 Rome, Italy; cesare.gruber@inmi.it (C.E.M.G.); fabio.tucci@inmi.it (F.G.T.); martina.rueca@inmi.it (M.R.); giulia.gramigna@inmi.it (G.G.); giulia.berno@inmi.it (G.B.); emanuela.giombini@inmi.it (E.G.); fabrizio.maggi@inmi.it (F.M.); 2Clinical and Research Infectious Diseases Department, National Institute for Infectious Diseases “Lazzaro Spallanzani” (IRCCS), 00149 Rome, Italy; valentina.mazzotta@inmi.it (V.M.); alessandra.vergori@inmi.it (A.V.); emanuele.nicastri@inmi.it (E.N.); andrea.antinori@inmi.it (A.A.); 3Laboratory of Microbiology, National Institute for Infectious Diseases “Lazzaro Spallanzani” (IRCCS), 00149 Rome, Italy; ornella.butera@inmi.it; 4North-Western Tuscany Blood Bank, Pisa University Hospital, 56124 Pisa, Italy; daniele.focosi@gmail.com; 5Department for Innovation in Biological, Agro-Food and Forest Systems (DIBAF), University of Tuscia, Via S. Camillo de Lellis s.n.c, 01100 Viterbo, Italy; ingrid.prandi@unitus.it (I.G.P.); gchillemi@unitus.it (G.C.); 6General Direction, National Institute for Infectious Diseases “Lazzaro Spallanzani” (IRCCS), 00149 Rome, Italy; francesco.vaia@inmi.it; 7Scientific Direction, National Institute for Infectious Diseases “Lazzaro Spallanzani” (IRCCS), 00149 Rome, Italy; enrico.girardi@inmi.it

**Keywords:** tixagevimab, cilgavimab, Spike, mAbs, SARS-CoV-2, mutations, quasispecies, deletion

## Abstract

Mutations in the SARS-CoV-2 Spike glycoprotein can affect monoclonal antibody efficacy. Recent findings report the occurrence of resistant mutations in immunocompromised patients after tixagevimab/cilgavimab treatment. More recently, the Food and Drug Agency revoked the authorization for tixagevimab/cilgavimab, while this monoclonal antibody cocktail is currently recommended by the European Medical Agency. We retrospectively reviewed 22 immunocompetent patients at high risk for disease progression who received intramuscular tixagevimab/cilgavimab as early COVID-19 treatment and presented a prolonged high viral load. Complete SARS-CoV-2 genome sequences were obtained for a deep investigation of mutation frequencies in Spike protein before and during treatment. At seven days, only one patient showed evidence of treatment-emergent cilgavimab resistance. Quasispecies analysis revealed two different deletions on the Spike protein (S:del138–144 or S:del141–145) in combination with the resistance S:K444N mutation. The structural and dynamic impact of the two quasispecies was characterized by using molecular dynamics simulations, showing the conservation of the principal functional movements in the mutated systems and their capabilities to alter the structure and dynamics of the RBD, responsible for the interaction with the ACE2 human receptor. Our study underlines the importance of prompting an early virological investigation to prevent drug resistance or clinical failures in immunocompetent patients.

## 1. Introduction

The long-acting anti-Spike (S) monoclonal antibody (mAb) cocktail tixagevimab/cilgavimab has been authorized for both pre-exposure prophylaxis (PrEP) and early treatment of COVID-19 by the U.S. Food and Drug Administration (FDA) on 8 December 2021 and by the European Medical Agency (EMA) on 25 March 2022 [1,2]. The exact Spike aminoacidic substitutions that confer resistance to each of the components have been recently identified. Mutations causing >30-fold increases in neutralization titers include R346X, K444X, and V445X for cilgavimab and F486X for tixagevimab. Such mutations can be constitutional in several Omicron sub-lineages or develop after treatment. In vitro, cilgavimab-selected variants that expressed Spike protein amino acid substitutions R346G/I, K444E/N/Q/R/T, or N450D were each associated with a >200-fold reduction in susceptibility [3]. The number of articles reporting the effects of tixagevimab/cilgavimab on clinical outcomes in patients with COVID-19 is limited [4,5,6]. In the first genomic surveillance study on tixagevimab/cilgavimab 300/300 mg treatment, Vellas et al. found resistance mutations R346X, K444X, and L452R in 73% (9/11) of BA.2-infected solid organ transplant recipients receiving heavy immunosuppression who were followed for up to 14 days [4]. Moreover, genomic characterization analysis performed by Ordaya et al. revealed K444N, G446D, L452X, F486V and Q493R substitutions occurring in SARS-CoV-2 Omicron subvariants and associated with the COVID-19 breakthrough in immunocompromised patients who received pre-exposure prophylaxis with tixagevimab/cilgavimab 300/300 mg [5].

Omicron BA.4/5 and their sub-lineages (BA.4.*/5.*) have baseline in vitro resistance to tixagevimab due to S:F486V [7], making cilgavimab resilience critical for drug activity (a negligible 5-fold reduction in activity occurs at baseline due to S:L452R). Zhang et al. showed through molecular dynamic simulations that the two Omicron variants can reduce the electrostatic attraction and impair the interaction between the receptor binding domain (RBD) and the complementary determining regions of the mAbs tixagevimab/cilgavimab [8].

Recently, the FDA revised the Emergency Use Authorization for tixagevimab/cilgavimab because of their expected reduced activity against actual circulating SARS-CoV-2 lineages [1]. However, some studies suggested that increasing the dose up to 600 mg may provide therapeutic benefit against less susceptible SARS-CoV-2 lineages and may ensure increased levels of protection for immunocompromised individuals [9,10]. The latest EMA release confirms tixagevimab/cilgavimab authorization for use in the EU. Moreover, the safety profile of this mAb is favorable despite generally mild side effects [2].

Here, we tested baseline and treatment-emergent resistance in patients at high risk of COVID-19 progression during the BA.4/5 wave in Italy. We retrospectively reviewed 22 consecutive outpatients at high risk for disease progression, as defined by Italian Drug Agency (AIFA) criteria, who received intramuscular tixagevimab/cilgavimab 300/300 mg as early treatment for COVID-19 at the National Institute for Infectious Diseases in Rome from April to December 2022.

## 2. Materials and Methods

The study group consisted of 22 randomly selected patients (14 females and 8 males; mean age of 69 ± 10 years and range of 54 to 85 years) with SARS-CoV-2 infection. All patients, prior to initiation of the study, had received no previous PrEP with tixagevimab/cilgavimab or other previous or concurrent COVID-19 antiviral treatment. A total of 21 patients were vaccinated: 16 patients with 3 doses and 5 patients with 4 doses of wild-type derived vaccine. One patient only received no dose of any vaccine. The treatment choice was based on clinical judgment. All patients were intramuscularly administered 300/300 mg of tixagevimab/cilgavimab. After informed consent, nasopharyngeal swabs (NPSs) were collected from all patients at day 0 and 7 of treatment. In a subgroup of 7 patients, an NPS was also collected at day 30 of therapy. The presence of SARS-CoV-2 RNA was detected by commercial RT-PCR assays. NPSs collected from all patients were analyzed by using the Alinity m SARS-CoV-2 Assay (Abbott, Chicago, IL, USA) targeting the RdRp and N genes of the viral genome. Whole genome sequencing was performed on available residual NPS samples. Nucleic acid extraction was performed by using the QiaSymphony automatic extractor (QIAGEN, Hilden, Germany) using the DSP Virus/Pathogen Kit (QIAGEN) starting at 600 μL and eluting in 70 μL of AVE buffer. Sequencing libraries were prepared using the Ion AmpliSeq SARS-CoV-2 Insight Research Assay, and next-generation sequencing (NGS) was carried out on the Ion Torrent Gene Studio S5 Prime (GSS5 Prime) platform following the manufacturer’s instructions (ThermoFisher, Waltham, MA, USA). Ethical approval for sequence analysis was obtained (no. 214/2020).

Raw reads with a mean quality Phred score greater than 20 were collected and trimmed using Trimmo-matic software v.0.36 [11]. SARS-CoV-2 whole genome assemblies were obtained using the easy-to-use SARS-CoV-2 assembler pipeline (ESCA), a novel reference-based genome assembly bioinformatic workflow that was specifically designed for SARS-CoV-2 data analysis [12]. For consensus sequence and mutation frequency assessment, whole genome sequencing was performed in parallel using a second assembler software RECoVERY v.3.4 (Rome, Italy) [13], and the Geneious Prime v.2019.2.3 (Biomatters Ltd, Auckland, New Zealand) program was adopted for comparing assemblies and mutational patterns.

### 2.1. In Silico Modeling of the Quasispecies

To evaluate the structural stability of mutated Spike proteins that were observed in vivo, structural models were generated through homology modeling using the SARS-CoV-2 Wuhan wild-type (WT) Spike structure (accession number: P0DTC2) as a template. Five computational systems were modeled as follows: wild-type system (WT_T0): Pt11 T0 (BA.4); QS1: WT_T0 model containing the 141–145 deletion; QS2: WT_T0 containing the 138–144 deletion; QS3: WT_T0 containing the 141–145 deletion and the K444N mutation; and QS4: WT_T0 containing the 138–144 deletion and the K444N mutation. For the in silico construction of the five glycosylated models of the Spike protein trimers, the cryo-EM structure (identification entry 6VYB [14]) available on the RCSD PDB website was used as a template. In this structure, it is important to note that monomer 2 (mon2) is present in the “up” conformation, i.e., favorable for interaction with the human cell receptor ACE2. The models were constructed through homology modeling using the SWISS-MODEL web application [15]. Then, the corresponding glycans of each monomer were inserted into the systems via the GLYCAM web server (www.glycam.org, accessed on 11 August 2023) following the well-documented description by Borocci et al. [16].

### 2.2. Molecular Dynamics Protocol

All molecular dynamics simulations were performed on the HPC Marconi-100 in Cineca, Italy, using the GROMACS 2020.2 software [17]. The Amber14SB [18], GLYCAM06 [19], and TIP3P [20] force fields were used to describe proteins, glycans, and water, respectively. For the glycosylated protein, the Spike–glycan bonds were first set, and then the topology was constructed using the AmberTools program [21]. The topologies were then converted to the GROMACS format using the ACPYPE script [22]. Before solvation, the systems were equilibrated in vacuum by applying a harmonic potential of 1000 KJ mol^-1^ nm^−2^ to the protein atoms to equilibrate the new glycan–protein bonds. The systems were then properly solvated and neutralized. A new minimization step was performed, maintaining the same constraint on the protein backbone atoms. After the system equilibration step where the constraints were removed, the 500 ns production phase of the molecular dynamics simulations was started. The simulations were performed with a time step of 2 fs, made possible by the H-bond constraint imposed by the P-LINCS algorithm [23]. The geometry of the water molecules was imposed by the SETTLE algorithm [24]. The electrostatic simulation scheme chosen was PME [25]. The production phase of all systems was performed in the NPT ensemble, with temperature and pressure controlled by the V-rescale thermostat [26] and the Parrinello-Ramhan barostat [27], respectively, at values of 300 K and 1 bar. This protocol has been described several times in the literature and used for other variants [16,28,29,30].

### 2.3. MD Analyses

To ensure the equilibration of the systems, all analyses were performed starting from 50 ns of the MDs production phase. The rmsf tool of GROMACS was used to obtain the data for this analysis. For the essential dynamics analyses, we considered only the c-alpha of the protein chain to obtain the dominant motion of the systems through the “covar” and “anaeig” tools present in the GROMACS package version 2020.6 on the concatenated trimeric trajectory [31]. This procedure has been documented multiple times in the existing literature [16,29]. To maintain the consistency of the WT_T0 numbering, we inserted dummy residues at the regions of the QS deletions. In order to extract representative frames from the MD simulations, we performed a cluster analysis of the trajectory using the GROMACS cluster tool. The centroid of the first cluster was chosen as the representative structure at 341.5 ns simulation time.

## 3. Results

### 3.1. Mutational Pattern Analysis

Cycle threshold (Ct) values significantly increased for all patients from day 0 to day 7 of tixagevimab/cilgavimab (Figure 1A), as well as Spike-specific immune responses (Figure 1B). Table 1 shows that all cases were infected by BA.4.*/5.*: three cases (i.e., cases no. 2, no. 18, and no. 19) were excluded from the 7-day sequencing because of the baseline occurrence of the cilgavimab-resisting mutations R346T or K444N/T. Of interest, the NPS at day 30 (available only for 2 of these 3 cases) was still positive (Table 1). None of the 19 patients was negative on day 7, while 3 out of 5 patients with an NPS available on day 30 were negative (Ct > 40).

Only 1 out of 19 cases (5.3%), which was the single unvaccinated member of the cohort, showed evidence of treatment-emergent virological resistance to cilgavimab. Specifically, case no. 11 showed a de novo S:K444N mutation (due to G22894T) at day 7 (53% prevalence of the quasispecies) associated with an increase in the RT-PCR cycle threshold from 16 to 22. Moreover, the same case showed the co-presence of two quasispecies with two different deletions on the same Spike protein domain: one deletion at position 21,972–21,992 (S:del138–144, 21 nt length, 57% prevalence of the quasispecies) and another deletion at position 21,982–21,996 (S:del141–145, 15 nt length, 43% prevalence of the quasispecies). Unfortunately, the patient was lost at follow-up, and additional data on viral shedding are unavailable. Investigations on the structural and dynamic combined effect of S:K444N and S:del138–144 or S:del141–145, particularly on the receptor-binding domain (RBD) region, were also performed by using different molecular dynamics simulations, each 500 ns long.

### 3.2. Structural Analysis

To evaluate the structural stability of the K444N in combination with the two observed quasispecies, S:del138–144 and S:del141–145, we built five computational models (see Section 2.1) named WT_T0: QS1, containing the 141–145 deletion; QS2, containing the 138–144 deletion; QS3, containing the 141–145 deletion and the K444N mutation; and QS4, containing the 138–144 deletion and the K444N mutation. The S trimer models in glycosylated form were then simulated for 500 ns of MD simulations, following our previous procedure [16,29].

During classical simulations, all systems were stable, indicating that the S protein could withstand the observed large deletions of consecutive residues without losing its stability.

Notably, the residue fluctuation analysis (RMSF) of the WT system indicated that the region where deletions occurred, the N-terminal domain (NTD), is quite flexible, especially in the conformation adopted by monomer 1 (mon1, black in Figure 2). Note that in our model, monomer 2 has the RBD in the up conformation, ready for interaction with the human ACE2 receptor.

Residues in the delete region, therefore, are in a flexible loop region (highlighted in blue in the insert of Figure 3) with no defined secondary structure. It is reasonable, therefore, to hypothesize that the flexible nature of the loop allows the S protein to tolerate the deletions without major functional effects on the protein while slightly changing the conformation and dynamics of RBD, likely reducing or abolishing the interaction with the antibodies.

To evaluate the structural and dynamic effects of these deletions, we performed essential dynamics (ED) analyses on the concatenated trajectories of the three monomers in the five simulated models—a type of analysis that has proven very useful in highlighting long-range correlated motions in the S protein [16,29,32]. ED analysis, in fact, is a principal component analysis, which allows one to separate the large collective protein movements connected to functional properties from the small, uninteresting motions [31] and, in the case of the S protein, catch the important up-and-down movement of the monomers, which is crucial for the interaction of the Spike with the ACE2 cell receptor.

Figure 4 shows the ED-filtered trajectories along eigenvector 1 (the one essential motion containing the great majority of the protein motions) for the five simulated systems. The movement along eigenvector 1, as expected, was dominated by the RBD up/down (see also Figure 5A,C). The overall structural flexibility of the S protein was maintained even in cases of deletion of residues in the 138–145 region, however a greater rigidity for RBD up/down movement in QS2/QS4 is observed (seen also in Figure 5).

The effect of the mutations on the protein motion along the first eigenvector can be better appreciated by looking at the per-residue fluctuations in the three systems WT_T0/QS1/QS3 and WT_T0/QS2/QS4 (Figure 5, panels A/B and C/D, respectively). By subtracting the fluctuations of the QS from the WT_T0, we observed a long-range effect of K444N on the RBD in the QS3 system (note the negative peak in residue 479in Figure 5B),in the fusion peptide proximal region (FPPR; residues 834–853), which has been proposed to play an important role in the transition between up and down conformations of the RBD [33], and in residue 628 (note the positive peak in Figure 5B). An effect on the same region was also present in the QS1 system, even with the opposite effect of increasing the fluctuation of this region as compared to WT_T0 (see the negative peak in residue 637 in Figure 5B).

The effect of the 138–144 deletion was observed in the QS2/QS4 systems (Figure 5C,D), with negative peaks in residue 257 and 847 and a positive peak in the RBD, residue 483, showing a higher rigidity of the up/down movement of this domain for QS2/QS4 compared to the WT_T0 (in line with Figure 4). (see Figure 5D).

## 4. Discussion

Our data, compared with those of Vellas et al. [4] (despite being about different Omicron sub-lineages and baseline resistances), demonstrate that anti-Spike mAb immune escape is more commonly detected in severely immunocompromised patients (9/11) than in our cohort (1/19), in which only 2 patients were solid organ transplant recipients (SOTRs).

Anti-Spike mAb resistance has been previously described for all authorized mAbs [34,35,36,37], including in patients treated with fully susceptible mAbs at baseline [38,39]. Unfortunately, the R346 and K444 residues have been described as sites of convergent evolution since summer 2022 [40], with widespread baseline in vitro resistance to cilgavimab. In this study, we observed, besides the known K444N mutation, the presence of a quasispecies characterized by two different deletions located in an unstructured loop in the S NTD. Structural and dynamic characterizations showed the conservation of the principal functional movements in the mutated systems and their capabilities to alter the structure and dynamics of the RBD, responsible for the interaction with the ACE2 human receptor.

In addition, our structural results on the observed S variants are in line with recent findings showing that S can adapt and remain functional with either a large number of mild mutations (as in the case of the Omicron variant) or a smaller number of more critical and disruptive mutations (as in the case of the Delta variant) [29].

Based on the baseline immune evasion of newly emerging SARS-CoV-2 Omicron subvariants, on 26 January 2023, the FDA revoked the emergency authorization for PrEP use of tixagevimab/cilgavimab mAbs [1]. Nonetheless, tixagevimab/cilgavimab is still authorized as PrEP or early treatment in many countries.

This study has potential limitations. First, the old age of most patients (19 out of 22 patients aged more than 65 years) could limit to whom the findings can be generalized. Second, the small sample size could influence research findings, making the conclusions narrow. Further studies collecting more patients with a larger spectrum of ages are then needed to make the findings more general.

## 5. Conclusions

Clinical or virological failures, especially in severely immunocompromised patients, should prompt extensive virological investigations to identify baseline in vitro resistance to mAbs early and to rule out, albeit uncommon, treatment-emergent resistance. Nonetheless, if new sub-lineages fully susceptible to tixagevimab/cilgavimab return prevalent, virological, or clinical failures in both cohorts of immunocompromised or immunocompetent patients, this should prompt early virological investigations to rule out treatment-emergent resistance. From the structural point of view, the S protein is capable of accommodating large deletions on consecutive residues without significant functional impairments, particularly in the 138–145 region (a flexible loop with no well-defined secondary structure). However, the presence of the 138–144 deletion in QS2 and QS4 seems to exert some influence on the flexibility of the RBD on Spike protein. Overall, these findings reveal the S protein’s structural resilience, opening the way for further exploratory research on the functionality of this class of viruses.

## Figures and Tables

**Figure 1 biomolecules-13-01538-f001:**
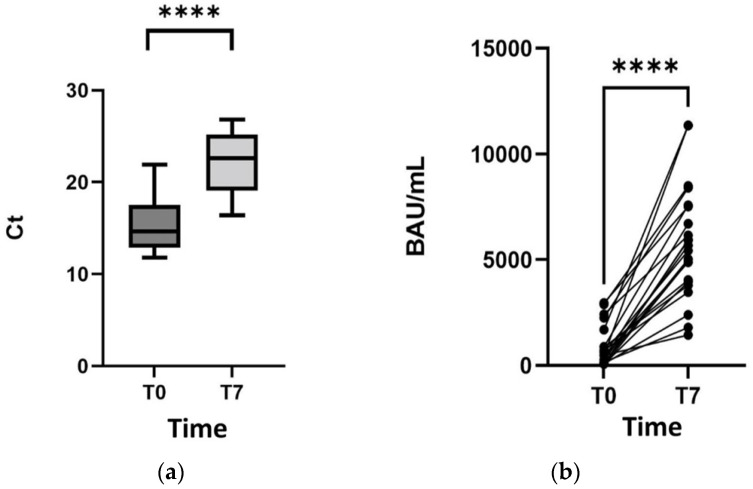
Cycle threshold values (**a**) and Spike-specific immune response (**b**) for all patients from day 0 to day 7 of tixagevimab/cilgavimab treatment. ********
*p* ≤ 0.0001.

**Figure 2 biomolecules-13-01538-f002:**
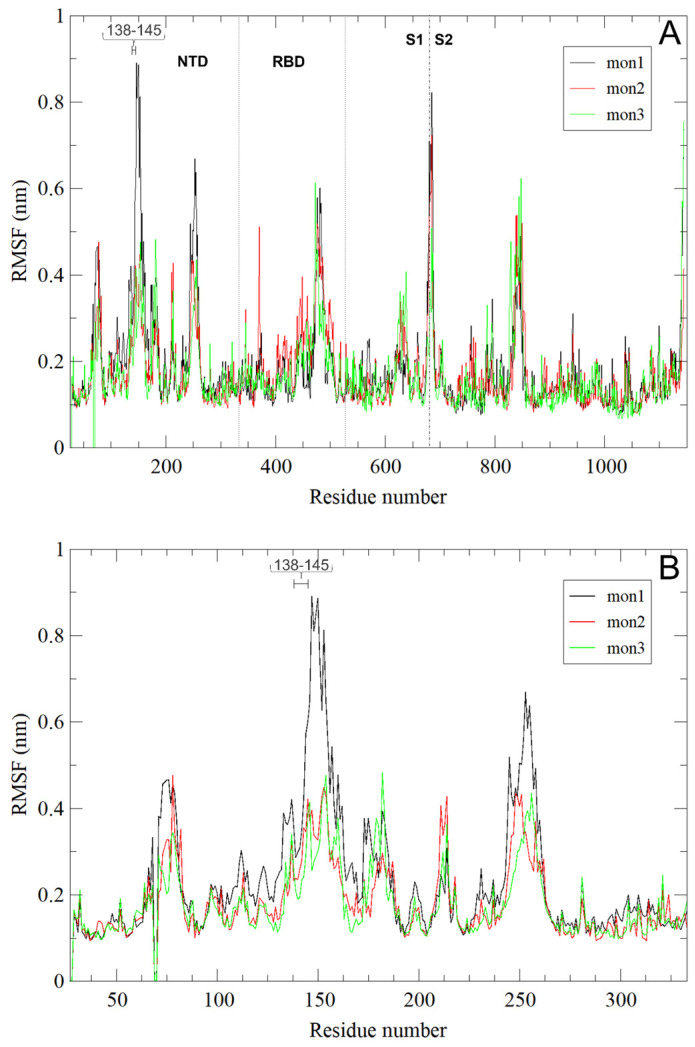
(**A**) Per-residue RMSF of SARS-CoV-2 WT_T0 (BA.4) S trimer. Spike monomers 1, 2, and 3 are colored black, red, and green, respectively. Monomer 2 is in the up conformation (see Materials and Methods), ready for interaction with the ACE2 human receptor. Dotted lines indicate the NTD and RBD regions. The dashed line at residue 681 highlights the S1/S2 boundary. Note that, to allow the comparison among different S variants, the residue numbers are reported as WT (Wuhan) equivalents, not considering the deletion and insertion of residues in the mutated systems. The region of residues 138–145, where the two deletions in the quasispecies systems are located, is highlighted with a horizontal line; (**B**) as in panel (**A**) for the NTD domain.

**Figure 3 biomolecules-13-01538-f003:**
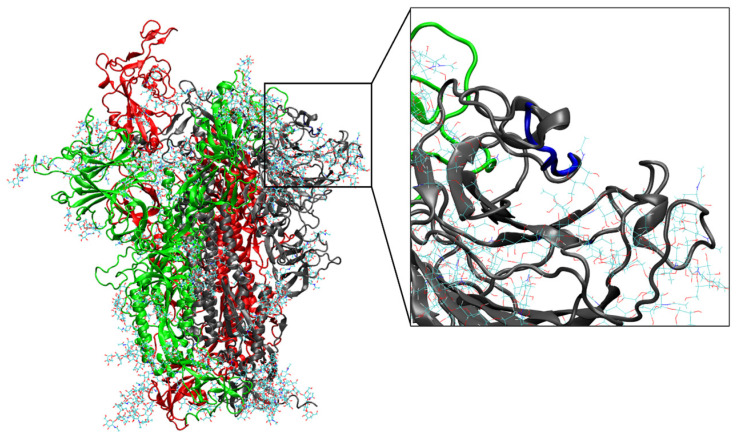
Cartoon representation of the representative frame (341.5 ns of MD simulation) for the WT_T0 (BA.4) S trimer. Spike monomers 1, 2, and 3 are colored black, red, and green, respectively. Residues 138–145, where the two deletions in the quasispecies systems are located, are highlighted in blue in the insert for monomer 1, which showed the highest RMSF values.

**Figure 4 biomolecules-13-01538-f004:**
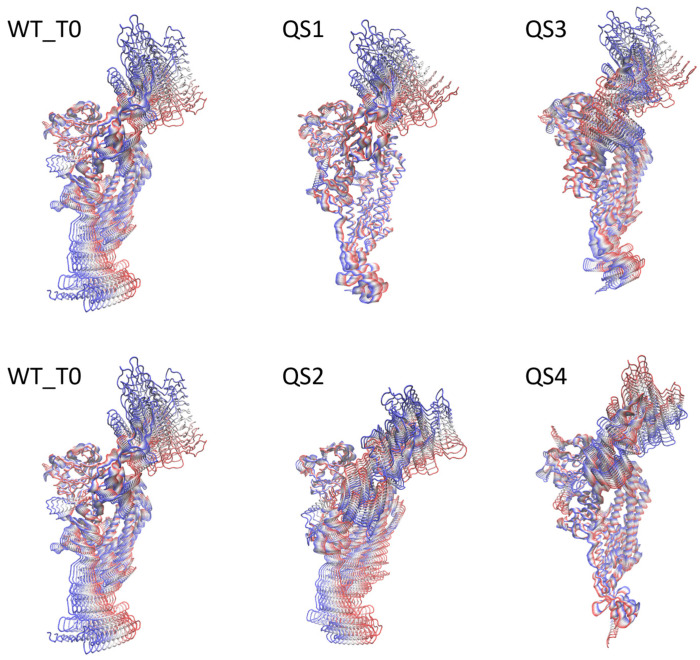
Projections of the essential dynamic movement along eigenvector 1 for the five simulated systems. Ten frames are represented, colored from red to blue. The WT_T0 system is reported twice to facilitate the comparison.

**Figure 5 biomolecules-13-01538-f005:**
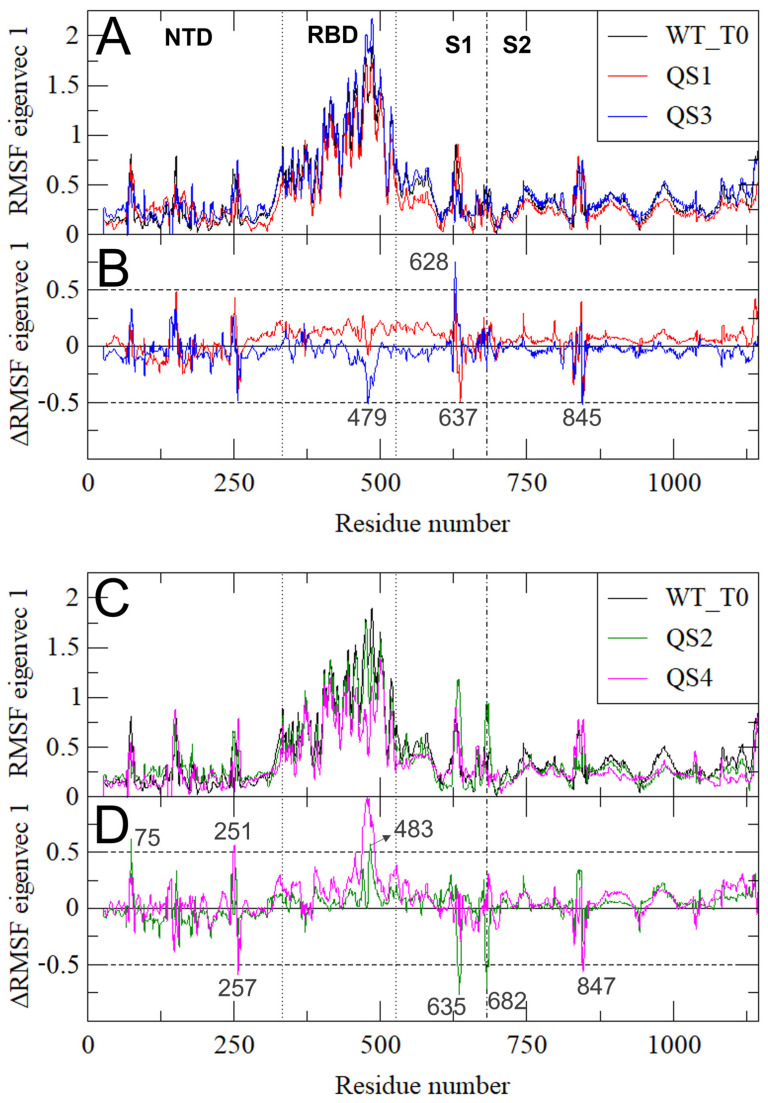
RMSF of the S-filtered trajectory along the first eigenvector. (**A**) WT_T0/QS1/QS3 system; (**B**) the differences between the filtered RMSF along the first eigenvector of WT_T0 and QS1/QS3. Residues above the threshold of |± 0.5 nm| (dashed lines) are indicated. Panels (**C**,**D**) as for (**A**,**B**) for WT_T0/QS2/QS4 systems.

**Table 1 biomolecules-13-01538-t001:** Baseline characteristics of tixagevimab/cilgavimab-treated patients and occurrence of treatment-emergent resistance mutations.

Pt	Gender	Age	Risk Factors for COVID-19 Progression	Immunodeficiency	SARS-CoV-2 Vaccination Status	SARS-CoV-2 PCR Cycle Threshold (Ct) on the Day of Cilgavimab Infusion			SARS-CoV-2 Lineage	Resistance Mutation (% in Quasispecies)	
						0	7	30		at Day 0	at Day 7
1	Female	73	age > 65 yrs, chronic lung disease	No	3 doses	11.9	21.2	Neg	BA.5.2.1	-	-
2	Male	68	age > 65 yrs, cardiovascular disease	No	3 doses	14.1	25.1	37.0	BA.5.2	K444N (47%)	-
3	Female	69	age > 65 yrs, cancer	Yes	3 doses	14.4	23.0	NA	BA.4	-	-
4	Female	72	age > 65 yrs, cancer	Yes	3 doses	12.9	19.4	36.9	BA.5.1	-	-
5	Male	60	chronic lung disease, obesity	No	3 doses	17.3	24.6	NA	BA.5.2.1	-	-
6	Female	84	age > 65 yrs, cancer, cardiovascular disease	Yes	4 doses	11.8	19.0	Neg	BA.5.1	-	-
7	Female	82	age > 65 yrs, obesity	No	3 doses	17.2	17.8	NA	BF.5 (BA.5.1.2.5) *	-	-
8	Female	55	diabetes, cardiovascular disease	No	3 doses	17.5	25.5	NA	BA.5.1.23	-	-
9	Male	70	age > 65 yrs, cardiovascular disease	No	3 doses	12.8	26.5	36.6	BA.5.2.1	-	-
10	Female	54	cancer	Yes	3 doses	21.9	25.7	Neg	BA.5.2	-	-
11	Male	63	neurologic disease	No	Unvaccinated	16.1	22.8	NA	BA.4	-	K444N (53%)
12	Female	74	age > 65 yrs, hematologic disease	Yes	4 doses	14.9	20.2	NA	BA.5.6	-	-
13	Male	81	age > 65 yrs, diabetes, chronic lung disease	No	3 doses	13.8	17.4	NA	BA.5.1	-	-
14	Female	76	age > 65 yrs, diabetes, cardiovascular disease, neurologic disease	No	3 doses	18.0	26.2	NA	BA.5.1.25	-	-
15	Female	76	age > 65 yrs, cardiovascular disease	No	3 doses	19.0	22.0	NA	BA.5.1	-	-
16	Male	85	age > 65 yrs, cardiovascular disease	No	4 doses	13.6	18.4	NA	BA.5.1.23	-	-
17	Female	55	cancer	Yes	3 doses	17.6	22.5	NA	BA.5.1	-	-
18	Male	63	renal impairment	No	3 doses	13.3	26.8	NA	BQ.1.1 (BA.5.3.1–1.1.1.1.1) *	R346T (100%) and K444T (100%)	-
19	Female	75	age > 65 yrs, diabetes, cardiovascular disease	No	3 doses	16.2	22.8	35.6	BA.4.6	R346T (100%)	-
20	Female	60	kidney transplantation	Yes	4 doses	12.3	19.1	NA	BA.5.2.1	-	-
21	Female	62	Sjögren syndrome, immunosuppressive therapies	Yes	3 doses	12.8	16.4	NA	BA.5.2.1	-	-
22	Male	61	kidney transplantation	Yes	4 doses	17.6	22.7	NA	BA.5.1	-	-

* Alias of BA.5 sub-lineage. NA: not available. Neg: negative. PCR: polymerase chain reaction.

## Data Availability

All data generated in this study are available upon reasonable request to the corresponding author.

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
