# Peer review of "Treatment-Emergent Cilgavimab Resistance Was Uncommon in Vaccinated Omicron BA.4/5 Outpatients"

_biomolecules, 2023, doi:10.3390/biom13101538_

Round 1

Reviewer 1 Report

Reviewer’s Comments:

The manuscript “Treatment-Emergent Cilgavimab Resistance Was Uncommon In Vaccinated Omicron BA.4/5 Outpatients” is very interesting work. Mutations in the SARS-CoV-2 Spike glycoprotein can affect monoclonal antibody efficacy. We retrospectively reviewed 22 immunocompetent patients at high risk for disease progression who received intramuscular tixagevimab/cilgavimab as early COVID-19 treatment and presented a prolonged high viral load. Complete SARS-CoV-2 genome sequences were obtained for a deep investigation of mutation frequencies in Spike protein before and during treatment. At seven days only 1 patient showed evidences of treatment emergent cilgavimab resistance (S:K444N). The structural and dynamic impact of two quasispecies with two different deletions on the Spike protein (S:del138-144 or S:del141-145), in combination with the S:K444N mutation, was characterized by molecular dynamics simulations.  However, the following issues should be carefully treated before publication.

1. In abstract, the author should add more scientific findings.

2. Keywords: the synthesized system is missing in the keywords. So, modify the keywords.

3. In the introduction part, the introduction part is not well organized and cited references should cite recently published articles such as 10.3390/molecules27196457, 10.3389/fchem.2022.1023316

4. Introduction part is not impressive and systematic. In the introduction part, the authors should elaborate the scientific issues in the Vaccinated Omicron research.

5. Results…, The author should provide reason about this statement “The residue numbers are reported as WT (Wuhan) equivalents, not considering the deletion and insertion of residues in the mutated systems”.

6. The authors should explain regarding the recent literature why “This analysis allows to separate the large collective protein movements connected to functional properties from the small, uninteresting motions”.

7. The author should explain the latest literature “Figure 4 shows the ED-filtered trajectories along eigenvector 1 for the five simulated systems”.

9. Comparison of the present results with other similar findings in the literature should be discussed in more detail. This is necessary to place this work together with other work in the field and to give more credibility to the present results.

10. The conclusion part is very weak. Improve by adding the results of your studies.

Minor editing of English language required

Reviewer 2 Report

Journal: Biomolecules

Title: Treatment-Emergent Cligavimab Resistance Was Uncommon in Vaccinated Omicron BA.4/5 Outpatients

The authors examined the treatment emergent resistance in patients at high risk of COVID-19 progression during the BA.4/5 wave in Italy. I would like to give the comments as below. 

Major Comments

1.     The authors should explain the methodology clearly despite of its laboratory process. Sampling strategy including recruitment method, sample type, sample points and its frequency, sample size calculation etc should be described in detail. Did the authors collect samples from all patients in three points (Day 0, 7 and 30)? 

2.     The authors described almost all received the COVID-19 vaccines, could you provide the information on type of vaccine including wild type derived or omicron derived vaccine? It is another interesting point on to examine any association to emergence of Cligavimab resistant strains by vaccine type. 

3.     The authors examine the treatment-Emergent Cligavimab Resistant strain among vaccinated COVID-19 confirmed patients. It is quite unclear whether all patients received any medicine during their COVID-19 infection or received any PrEP Tixagevimab/ Cilgavimab before their infection. If they received PrEP Tixagevimab/ Cilgavimab before infection, how long between PrEP and COVID-19 diagnosis. 

4.     Almost all patients were old age group (age>65: 19/22), the result is confined to the old age group, and considered to add in limitation. 

5.     The authors included those three patients already infected with targeted resistant strains at baseline (#2, 18 and 19), please explain the reason of including those three patients if the aim of this study is to examine the emergence of resistant strains. 

6.     The authors should consider the small sample size to conclude the findings. 7.     The authors should consider including the control group for this assessment to present the statistically significance of uncommonness of resistant strain emergence in vaccinated patients.  

Reviewer 3 Report

This is an analysis of mutations in the spike protein that confer resistance to monoclonal antibody treatment. Only one patient of the 22 patients exhibited a resistance mutation at day 7 and no resistance mutation at baseline (day 0). Three patients having a mutation at baseline were eliminated from the analysis. This was S:K444N mutation and the presence of a quasispecies with 2 deletions that were different in the S NTD were identified. The structural analysis was evaluated using a bioinformatic approach that was useful and explained the possible reason that the neutralization occurred despite the mutation. It would have been better to evaluate more mutations, but these weren’t available. It would be good to evaluate one other patient.

The data also showed that the cohort had only one patient that developed resistance to monoclonal antibodies against SARS-CoV-2 and he was not immunocompromised, whereas other reports with immunocompromised patients showed substantially more of these types of patients developed resistance to monoclonal antibody treatment through mutations in the Spike protein interacting with the monoclonal antibody. A larger cohort would have been better to evaluate the scarcity of these mutations.

RBD and NTD are not defined well. The deletion regions are well defined in the figures and the analysis, but the significance and function of these areas is not well described.

Round 2

Reviewer 2 Report

The authors responded well to the previous concerns. No comment is added.